# Effects of Surgical Treatment for Obstructive Sleep Apnea on Renal and Survival Outcomes in Patients with Chronic Kidney Disease: A Taiwanese Nationwide Cohort Study

**DOI:** 10.3390/jcm11154411

**Published:** 2022-07-29

**Authors:** Juen-Haur Hwang, Ben-Hui Yu, Yi-Chun Chen

**Affiliations:** 1Department of Otolaryngology-Head and Neck Surgery, Dalin Tzu Chi Hospital, Buddhist Tzu Chi Medical Foundation, Chiayi 622, Taiwan; juenhaur@gmail.com; 2School of Medicine, Tzu Chi University, Hualien 970, Taiwan; 3Department of Medical Research, China Medical University Hospital, China Medical University, Taichung 404, Taiwan; 4Department of Radiation Oncology, Dalin Tzu Chi Hospital, Buddhist Tzu Chi Medical Foundation, Chiayi 622, Taiwan; bhyu0418@gmail.com; 5Division of Nephrology, Department of Internal Medicine, Dalin Tzu Chi Hospital, Buddhist Tzu Chi Medical Foundation, Chiayi 622, Taiwan

**Keywords:** obstructive sleep apnea, surgical treatment, CKD, ESRD, mortality

## Abstract

The association between surgical treatment for obstructive sleep apnea (OSA) in chronic kidney disease (CKD) patients and end-stage renal disease (ESRD) and survival outcomes is not established, and this study aimed to evaluate this association. A retrospective cohort analysis was conducted from 2001 to 2015, including 32,220 eligible CKD patients with incident OSA. By 1:3 propensity score matching, 1078 CKD patients with incident OSA who received surgery (treated cohort) and 3234 untreated cohort who never received surgery were analyzed. The risk of ESRD in the competing mortality was significantly lower in the treated cohort than in the untreated cohort, with an adjusted hazard ratio (aHR) of 0.38 (95% confidence interval (CI0, 0.15–0.97; *p* = 0.043). In addition, the adjusted HRs of overall, cardiovascular, and non-cardiovascular mortality in the treated and untreated cohorts were 2.54 (95% CI, 1.79–3.59; *p* < 0.0001), 1.46 (95% CI, 0.29–7.22; *p* = 0.64), and 2.62 (95% CI, 1.83–3.75; *p* < 0.0001), respectively. Furthermore, the risks of overall and non-cardiovascular mortality for the treated cohort primarily occurred during a 3-month follow-up. In conclusion, surgical treatment for incident OSA in CKD patients was associated with decreased ESRD risk, but with increased non-cardiovascular mortality risk, especially within 3 months after surgical treatment.

## 1. Introduction

Sleep disturbances, including obstructive sleep apnea (OSA), are associated with or increase the risks of many diseases, including cardiometabolic disorders, tinnitus, neurodegenerative diseases, erectile dysfunction, and cancers, etc. [1,2,3,4,5,6]. Sleep disorders might also increase mortality risk [7]. OSA is a respiratory sleep disorder characterized by episodes of partial or complete upper airway collapse with reduction or complete cessation of airflow. Several mechanisms for OSA, such as intermittent hypoxemia, sleep disruption, and hypercapnia over the hypothalamic–pituitary–adrenal axis, have also been associated with poor neurocognitive performance [8].

A meta-analysis found that OSA and chronic kidney disease (CKD) are significantly associated, particularly in their more severe categories [9]. OSA was even recognized as a potential risk factor for the development and progression of CKD [10]. Compared to the general population, patients with CKD have a higher prevalence of OSA, central sleep apnea [11], and various sleep-related disorders [12]. Thus, a bidirectional relation between sleep apnea and CKD was found [11]. A significantly higher incident end-stage renal disease (ESRD) risk was observed for patients older than 40 years with OSA than for matched controls [13]. Patients with CKD have increased morbidity and mortality, mainly due to cardiovascular disease [11]. Furthermore, the presence of sleep apnea in the CKD population was also associated with an increased risk of cardiovascular events and mortality [11].

Body weight reduction in the case of obesity, continuous positive airway pressure (CPAP) therapy, oral appliances, and corrective surgery in the case of airway obstruction were mostly suggested to reverse the health risks of sleep apnea [8,10]. CPAP treatment seems to improve cognitive defects associated with OSA, but the effects of the other therapies on cognitive function were still inconclusive [8]. Recent studies suggested that CPAP treatment may be beneficial for renal function among patients with OSA [10]. However, it is still unclear whether or notsurgical treatment for OSA can decrease the development of ESRD and/or mortality in patients with CKD. Therefore, we aimed to investigate this issue through a large-scale cohort study.

## 2. Materials and Methods

### 2.1. Data Source

This retrospective cohort study analyzed claims data from Taiwan’s 2005 Longitudinal Generation Tracking Database (LGTD2005) between 1 January 2000 and 31 December 2016. The LGTD2005 has been well described in our prior research [14]. In brief, it is a de-identified national health insurance database and thus, this study did not require informed consent and was exempt from full review by the Institutional Review Board of Buddhist Dalin Tzu Chi Hospital (B10603017-1 and B10804003). It adopts the International Classification of Diseases, 9th and 10th Revision, Clinical Modification (ICD-9/10-CM) diagnosis codes to define diseases and contains comprehensive medical information of 2 million people randomly sampled from all beneficiaries in Taiwan’s National Health Insurance (NHI) program. Thus, there was no significant difference in age, gender, region, ambulatory care, and inpatient expenditures between the LGTD2005 and the compulsory universal NHI program that covers more than 99% of all legal residents of Taiwan [14].

### 2.2. Study Population

We first selected 376,809 patients with a diagnosis of CKD according to ICD-9-CM codes [14,15,16,17] between 1 January 2001 and 31 December 2015 from the outpatient and inpatient claims (Figure 1). The exact CKD stage could not be assessed because of a lack of laboratory information in the NHI claims data [14,15]. We excluded 19,510 CKD patients who were aged < 18 years, had missing data, had a diagnosis of ESRD or OSA before the CKD inception date, and expired or dropped out before the CKD inception date; we obtained 357,299 CKD patients without OSA. We further identified 32,220 incident OSA patients after the CKD inception date and divided them into two groups according to exposure to surgical treatment for OSA, which was defined as ever admission due to OSA or the presence of ICD-9/10 surgical procedure codes for OSA [5]. CKD patients who ever received surgical treatments for OSA between 2001(1 year after the start of the LGTD2005 inclusion period) and 2015 (1 year before the end of the LGTD2005 inclusion period) were designated as the treated cohort (*n* = 1171). Those who were never treated with surgery for OSA between 2001 and 2015 were designated as the untreated cohort (*n* = 31,049). Indeed, other treatments, such as weight control or CPAP, could not be assessed in the NHI claims data, which may exist both in the treated and untreated groups. The period was chosen to ensure that a minimum of 1 year of follow-up was available for each participant. Then, the propensity score was used to adjust for the baseline differences between the treated and untreated cohorts by using the logistic regression built on all covariates. The propensity score model provided fair discrimination between the two cohorts (c-index, 0.71). Finally, 1078 patients in the treated cohort and 3234 in the untreated cohort were subjected to analysis by 1:3 propensity score matching.

To prevent immortal bias, we used the incident user design (exposed surgical treatment for OSA), with follow-up for each treated patient beginning on the date of the first surgical treatment for OSA, and the matching method, in which matched untreated patients must have been alive when surgical treatment for OSA commenced. In this situation, cohort entry became the date of first surgical treatment for OSA, and any time between CKD inception date and first surgical treatment for OSA was not counted for either group [13]. The index date of the treated cohort was set at the day when surgical treatment for OSA commenced; namely, treated patients survived from the CKD inception date to this index date. Meanwhile, the qualified propensity-matched untreated patients should have remained at risk in the same time interval to avoid the immortal time bias. Their index dates were set at the end of the same time interval, namely, the corresponding day of treated patients.

### 2.3. Covariate

We identified age, sex, and baseline comorbidities (including diabetes, hypertension, coronary heart disease, hyperlipidemia, and chronic liver disease) within a year before the CKD inception date [14]. We considered medical visits within a year beforethe CKD inception date and the Deyo–Charlson comorbidity indexscore for the control of confounding factors in studies using administrativedatabases [14,16] and angiotensin-converting enzyme inhibitor (ACEI)/angiotensin II receptor antagonist (ARB) as a confounding drug [14,17] because of their renoprotection from CKD progression.

### 2.4. Study Outcomes

Both cohorts were followed from the index date to ESRD occurrence, death, or the end of 2016, whichever came first. The latter two were considered as censoring observations. Moreover, death before ESRD occurrence was consideredacompetingriskevent [16]. ESRD was identified in the Registry for Catastrophic Illness Patient Database [14,16,17], a subset of the LGTD2005. All Taiwanese patients who develop ESRD and require long-term dialysis can obtain a catastrophic illness certificate after a rigorous review by the NHI Administration and have no co-payments for health care. Thus, the diagnostic accuracy of ESRD is reliable. Death was defined as withdrawal from the high coverage of the NHI program [18].

### 2.5. Statistical Analyses

We compared baseline characteristics between both study cohorts based onthe two-sided *t*-test for continuous variables and the Chi-squared test for categorical variables. We applied the modified Kaplan–Meier method and Gray’s method [16,17,18] to calculate and compare the cumulative incidence of ESRD in data with competing risks and tested differences between the study cohorts using the modified log-rank test. After confirming the assumption of proportional hazards by plotting the graph of the survival function versus the survival time and the graph of the log [–log(survival)] versus the log of survival time, we performed the modified Coxproportional hazard model in the presence of competing mortality and Cox regression toexamine the association of surgical treatment for OSA with ESRD and mortality, respectively, with adjustment for all covariates listed in Table 1.

The cumulative incidences and risks of overall, cardiovascular, and non-cardiovascular mortality were also estimated. Cardiovascular mortality was defined as death attributable to any cardiovascular event including the heart, brain, or blood vessels. We further performed a subgroup analysis to explore the relationship between surgical treatments for OSA and mortality outcomes stratified by follow-up interval (0–3 months vs. more than 3 months). Data were analyzed using SAS (version 9.4; SAS Institute, Inc., Cary, NC, USA). A two-sided *p*-value less than 0.05 was considered statistically significant.

## 3. Results

### 3.1. Demographic Characteristics of the CKD with Incident OSA Cohort

The average age of both cohorts was 46 years, and 67% were men (Table 1). The proportions of all covariates were similar between both cohorts.

### 3.2. Fifteen-Year Cumulative Incidences and Risks of Study Outcomes

The mean duration of follow-up was 4.5 years for the treated cohort and 4.7 years for the untreated cohort (Table 2). Among the 4312 CKD patients with incident OSA, 46 patients (1.1%) developed ESRD during the study period, with 5 (0.5%) from the treated cohort and 41 (1.3%) from the untreated cohort. Therisk of ESRD was significantlylower in the treatedcohort (15-year cumulative incidence,0.76%; 95% confidence interval (CI), 0.26–1.85%) than in the untreated cohort (1.19%; 95% CI, 0.79–1.74%) (*p* = 0.037), with an adjusted hazard ratio (aHR) of 0.38 (95% CI, 0.15–0.97; *p* = 0.043). A total of 138 patients (3.2%) died during the study period, with 53 (4.9%) from the treated cohort and 85 (2.6%) from the untreated cohort. The 15-yearcumulative incidence of overall mortality was 5.57% (95% CI, 4.06–7.40%) and 2.54% (95% CI, 1.91–3.30%) in the treated and untreated cohorts, respectively (*p* < 0.0001). That of non-cardiovascular mortality was 5.29% (95% CI, 3.83–7.08%) and 2.26% (95% CI, 1.68–2.98%) in the treated and untreated cohorts, respectively (*p* < 0.0001). In addition, the risks of overall mortality (aHR, 2.54; 95% CI, 1.79–3.59, *p* < 0.0001) and non-cardiovascular mortality (aHR, 2.62; 95% CI, 1.83–3.75, *p* < 0.0001) were significantly higher in the treatedcohortthan in the untreated cohort. However, the risk of cardiovascular mortality was not significantly different in both cohorts.

### 3.3. Subgroup Analysis of Mortality Outcome Stratified by Follow-Up Interval

In a subgroup analysis (Table 3), the risks of overall mortality (aHR, 8.81, *p* = 0.0003 vs. 2.18, *p* = 0.0003) and non-cardiovascular mortality (aHR, 7.84, *p* = 0.0007 vs. 2.3, *p* < 0.0001) for the treated cohort, compared with the untreated cohort, were more pronounced during the follow-up period of less than 3 months than that of more than 3 months. However, compared with the untreated cohort, the risk of cardiovascular mortality for the treated cohort was not significantly different during the follow-up period of less than 3 months and that of more than 3 months.

## 4. Discussion

Using incident user design, propensity score matching, and competing risk analysis, this large-scale cohort study demonstrated that CKD patients with incident OSA who received surgical treatment exhibited an association with a lower risk of ESRD by 62%, but 1.54-fold and 1.62-fold increased risks of overall and non-cardiovascular mortality, respectively. However, the cardiovascular mortality risk did not differ significantly whether incident OSA in CKD patients received surgical treatment or not. Further subgroup analysis by follow-up interval addressed the pronounced risks of overall and non-cardiovascular mortality in the treated cohort especially within 3 months after surgical treatment.

The overnight changes in serum and urine acute kidney injury (AKI) markers after apneic episodes during sleep reflected an increased risk of subclinical AKI in OSA patients [19]. Furthermore, OSA-related hypoxia could increase oxidative stress, inflammation, and sympathetic activation that collectively worsen the progression of renal disease. On the contrary, CKD can increase the severity of sleep apnea through uremic neuropathy and myopathy, altered chemosensitivity, and hypervolemia [20]. Thus, there was a vicious cycle between OSA and CKD and/or ESRD. Previous studies showed that fluid removal by ultrafiltration in ESRD markedly improved sleep apnea severity [21]. In turn, CPAP treatment may have a beneficial effect on renal function among patients with OSA [7]. However, Nowicki et al. [19] reported that CPAP therapy is not protective against AKI, but may reduce some of its risk factors, including high blood pressure and endothelial damage. Furthermore, the studies discussing surgical treatments on the risk of OSA-related diseases were limited.

Surgical treatment for OSA might improve quality of life (QOL) and/or decrease the severity of some OSA-related diseases. For example, respiratory disturbances and arousals were improved after upper airway surgery [22]. Upper airway surgery for OSA could improve the QOL, compared to using non-compliant CPAP devices or mandibular advancement splints [23]. Nasal obstruction relieving and uvulopalatopharyngoplasty surgical procedures could decrease the severity of erectile dysfunction in patients of OSA with erectile dysfunction [24,25]. Adenotonsillectomy for OSA may improve a child’s school performance and long-term cognitive ability [26].

Surgical treatment for OSA could reduce plasma TNF-alpha levels and decrease the risk of cardiovascular disease [27,28]. Moreover, surgical treatment may reduce the risk of primary brain cancers in subjects with OSA [3]. Recently, we found that surgical treatments could reduce the risk of erectile dysfunction in OSA patients by a large-scale cohort study [5]. This study showed that surgical treatment for OSA in CKD patients could decrease ESRD risk, but increase mortality risk, especially for non-cardiovascular mortality and within 3 months after surgical treatment. It is reasonable to see the decreased risk of ESRD in CKD patients who received surgical treatment for OSA because renal hypoxia and inflammation could be improved after surgery. However, it is somewhat strange to see the increased overall mortality risk paradoxically in the treated group. Therefore, we performed subgroup analysis additionally. We found that surgical treatment only increased non-cardiovascular mortality, but not cardiovascular mortality, especially within 3 months after surgery. According to a previous study, postoperative bleeding was significantly higher in OSA patients receiving uvulopalatopharyngoplasty alone than those receiving uvulopalatopharyngoplasty plus tongue/hypopharyngeal surgery (6.19% vs. 4.31%, *p* = 0.004) [29]. The incidences of complications and fatal rate were also higher in the multilevel surgery group (4.63% and 0.19%, respectively) than in the uvulopalatopharyngoplasty-only group (1.6% and 0.09%, respectively). There is a statistically significant correlation between the year of operation and the rate of complications, with an increased incidence of complications in recent years [30]. Thus, we supposed that the increased mortality might be related to surgical complications in patients with CKD.

This study had some weak points or limitations. First, the NHI claims data lack information on laboratory data (e.g., serum creatinine). Thus, we could not assess the exact CKD stage and further address the association between the CKD stage and the effect of surgical treatments for OSA. The method of adopting ICD diagnostic codes to identify CKD patients in Taiwan has been used in previous high-impact publications [14,15,16,17]. Moreover, the severity of OSA, body morphology, and other therapy for OSA (e.g., weight control, CPAP) also could not be assessed in the NHI claims data. Thus, the intergroup analysis of the severity of OSA was not presented in the current study. However, these unrecorded factors might violate the results equally in both cohorts. Thus, we supposed that our results would be similar as we reported. Second, the indication for surgical intervention was generally based on the willingness of OSA patients, but not by a randomized method or on the severity of OSA itself. We supposed that the severity of OSA in the treated and untreated groups was similar. Although the severity of OSA in the treated group tended to be more severe compared to the untreated group, its calculated incidence of ESRD was lower. Thus, these findings indeed reflected that surgical treatment for OSA in CKD patients could significantly reduce the incidence of ESRD in CKD patients. Third, the surgical methods for OSA could be assessed in the NHI claims data, but we had treated them collectively as one surgical group due to multilevel surgery being performed frequently in modern medicine. Fourth, some CKD patients with OSA in the non-surgical and surgical groups might receive CPAP therapy. It was possible that patients who underwent surgery were more willing to treat themselves and were more likely to use CPAP. Thus, the incidence of ESRD due to the difference in CPAP usage could not be ignored in this study. However, OSA patients who had received surgical treatment were less willing to use CPAP in Taiwan. Therefore, we supposed that the conclusion of this study would not differ. On the contrary, the effect of surgical treatment for OSA in CKD patients would be more significant because some subjects in the non-surgical group were not absolutely untreated. Fifth, the cut point for the follow-up interval of surgical outcomes was not standardized. One research assessed them as short-term (≤14 days), intermediate (15–60 days), and long-term (61–183 days) intervals [29]. However, we assessed the post-operative mortality as short-term (≤3 months) and long-term (>3 months) in this study.

## 5. Conclusions

Surgical treatment for incident OSA in CKD patients was associated with decreased ESRD risk, but with increased non-cardiovascular mortality risk, especially within 3 months after surgical treatment. Therefore, more attention should be paid to all CKD patients with OSA if they plan to receive surgical treatment for OSA in the future.

## Figures and Tables

**Figure 1 jcm-11-04411-f001:**
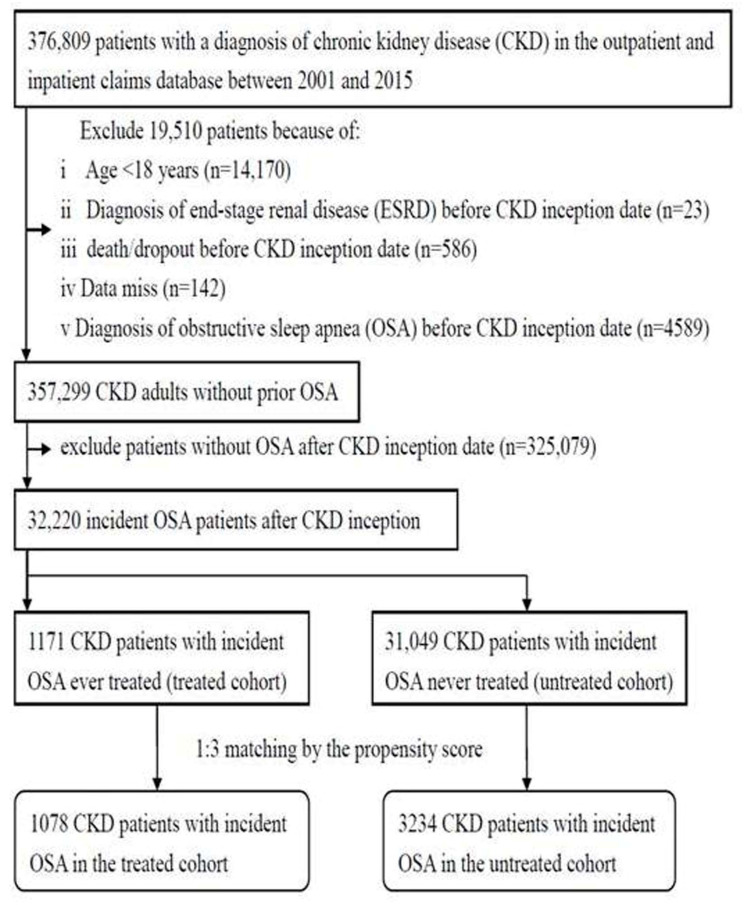
Flow diagram of the enrollment process.

**Table 1 jcm-11-04411-t001:** Baseline characteristics of CKD patients with incident obstructive sleep apnea (OSA) by the use of surgical treatment for OSA.

Variable	Propensity Score-Matched CKD Patients with Incident OSA (*n* = 4312)		*p*-Value
	Treated Cohort	Untreated Cohort	
	(*n* = 1078)	(*n* = 3234)	
Sex, *n* (%)			0.97
Men	719 (66.7)	2155 (66.6)	
Women	359 (33.3)	10797 (33.4)	
Age (year), *n* (%)			0.99
18–34	235 (21.8)	710 (22.0)	
35–44	272 (25.2)	806 (24.9)	
45–54	291 (27.0)	877 (27.1)	
≧55	280 (26.0)	841 (26.0)	
Mean (±SD)	46.1 ± 13.5	46.9 ± 14.6	0.10
Comorbidities, *n* (%)			
Diabetes	122 (11.3)	326 (10.1)	0.25
Hypertension	303 (28.1)	893 (27.6)	0.75
Coronary heart disease	106 (9.8)	2969 (9.22)	0.51
Hyperlipidemia	188 (17.4)	544 (16.8)	0.64
Chronic liver disease	111 (10.3)	394 (9.2)	0.24
Charlson comorbidity index, *n* (%)			0.89
0	5311 (49.3)	1592 (49.2)	
1	2324 (21.5)	728 (22.5)	
2	1779 (16.4)	516 (16.0)	
≧3	198 (12.8)	398 (12.3)	
Mean (±SD)	0.99 ± 1.25	0.98 ± 1.24	0.78
Number of medical visits, *n* (%)			0.99
1–12	345 (32.0)	1041 (32.2)	
13–24	338 (31.4)	1009 (31.2)	
≧25	395 (36.6)	1184 (36.6)	
Mean (±SD)	23.6 ± 19.3	23.0 ± 18.3	0.37
Confounding drug use, *n* (%)			
ACEI/ARB	200 (18.6)	565 (17.5)	0.42

Categorical variables given as number (percentage); continuous variable, as mean ± standard deviation (SD). Abbreviations: CKD, chronic kidney disease; ACEI/ARB, angiotensin–converting–enzyme inhibitor/angiotensin II receptor blocker.

**Table 2 jcm-11-04411-t002:** Study outcomes and incidences in the three study cohorts.

Outcomes	Treated Cohort (*n* = 1.078)	Untreated Cohort (*n* = 3.234)	*p*-Value	Adjusted HR (95% CI)	*p*-Value
ESRD				0.38 * (0.15–0.97)	0.043
Mean follow-up (±SD)	4.5 ± 3.6	4.7 ± 3.6			
Event (*n*, %)	5 (0.5)	41 (1.3)			
Cumulative incidence (%, 95% CI)	0.76% (0.26–1.85%)	1.19% (0.79–1.74%)	0.037		
Overall mortality				2.54 ^#^ (1.79–3.59)	<0.0001
Mean follow-up (±SD)	4.5 ± 3.6	4.7 ± 3.6			
Event (*n*, %)	53 (4.9)	85 (2.6)			
Cumulative incidence (%, 95% CI)	5.57% (4.06–7.40%)	2.54% (1.91–3.30%)	<0.0001		
Cardiovascular mortality				1.46 ^#^ (0.29–7.22)	0.64
Mean follow-up (±SD)	4.5 ± 3.6	4.7 ± 3.6			
Event (*n*, %)	2 (0.2)	7 (0.2)			
Cumulative incidence (%, 95% CI)	0.29% (0.06–1.05%)	0.28% (0.11–0.64%)	0.89		
Non-cardiovascular mortality				2.62 ^#^ (1.83–3.75)	<0.0001
Mean follow-up (±SD)	4.5 ± 3.6	4.7 ± 3.6			
Event (*n*, %)	51 (4.7)	78 (2.4)			
Cumulative incidence (%, 95% CI)	5.29% (3.83–7.08%)	2.26% (1.68–2.98%)	<0.0001		

Abbreviations: HR, hazard ratio; CI, confidence interval; ESRD, end-stage renal disease; SD, standard deviation. * Adjusted for all covariates (age per year, sex, comorbidity, number of medical visits, Charlson comorbidity index, and ACEI/ARB) and competing mortality. ^#^ Adjusted for all covariates (age per year, sex, comorbidity, medical visits, Charlson comorbidity index, and ACEI/ARB).

**Table 3 jcm-11-04411-t003:** Association between surgical treatment for obstructive sleep apnea and mortality outcomes stratified by follow-up interval.

Follow-Up Interval	Overall Mortality			Cardiovascular Mortality			Non-Cardiovascular Mortality		
	Event	Adjusted HR	*p*-Value	Event	Adjusted HR	*p*-Value	Event	Adjusted HR	*p*-Value
0–3 months									
Untreated (*n* = 3.234)	4	1 (Reference)		0	1 (Reference)		4	1 (Reference)	
Treated (*n* = 1.078)	10	8.81	0.0003	1	Not converged		9	7.84	0.0007
>3 months									
Untreated (*n* = 3.234)	81	1 (Reference)		7	1 (Reference)		74	1 (Reference)	
Treated (*n* = 1.078)	43	2.18	<0.0001	1	0.90	0.93	42	2.30	<0.0001

Abbreviations: HR, hazard ratio. Adjusted for all covariates (age per year, sex, comorbidity, number of medical visits, Charlson comorbidity index, and ACEI/ARB).

## Data Availability

Restrictions apply to the availability of these data. Data were obtained from the National Health Insurance database and are available from the authors with the permission of the National Health Insurance Administration of Taiwan.

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
