# Peer review of "Effects of Surgical Treatment for Obstructive Sleep Apnea on Renal and Survival Outcomes in Patients with Chronic Kidney Disease: A Taiwanese Nationwide Cohort Study"

_jcm, 2022, doi:10.3390/jcm11154411_

Round 1
Reviewer 1 Report
Introduction
- Patients with chronic kidney disease have increased morbidity and mortality, mainly due to cardiovascular disease. Compared with the general population, patients with chronic kidney disease have an increased prevalence of both OSA and central sleep apnea, and the presence of sleep apnea in this population has been associated with an increased risk of cardiovascular events and mortality. Although OSA can lead to an increase in the rate of kidney function decline, there is also evidence that the presence of end-stage renal disease can lead to worsening of sleep apnea, indicating a bidirectional relation between sleep apnea and chronic kidney disease., please discuss and cite doi:10.1016/j.chest.2019.09.004
- Obstructive Sleep Apnea (OSA) syndrome is a respiratory sleep disorder characterized by partial or complete episodes of upper airway collapse with reduction or complete cessation of airflow. Although the connection remains debated, several mechanisms such as intermittent hypoxemia, sleep deprivation, hypercapnia disruption of the hypothalamic-pituitary-adrenal axis have been associated with poor neurocognitive performance. Different treatments have been proposed to treat OSAS patients as continuous positive airway pressure (CPAP), mandibular advancement devices (MAD), surgery; however, the effect on neurocognitive functions is still debated. please cite doi:10.3390/bs11120180
Methods
- apply strobe guidelines to improve the structure
- add a flow diagram to clarify the protocol
Results
when you describe a comparison always report numbers and p value
- In an interesting study the pattern of overnight changes in serum and urine AKI markers after apneic episodes during sleep may suggest an increased risk of subclinical AKI in patients with OSA. The CPAP therapy is not protective against AKI, but may reduce some of its risk factors, including high blood pressure and endothelial damage. please discuss and cite doi:10.17219/acem/123356
Author Response
- -Patients with chronic kidney disease have increased morbidity and mortality, mainly due to cardiovascular disease. Compared with the general population, patients with chronic kidney disease have an increased prevalence of both OSA and central sleep apnea, and the presence of sleep apnea in this population has been associated with an increased risk of cardiovascular events and mortality. Although OSA can lead to an increase in the rate of kidney function decline, there is also evidence that the presence of end-stage renal disease can lead to worsening of sleep apnea, indicating a bidirectional relation between sleep apnea and chronic kidney disease., please discuss and cite doi:10.1016/j.chest.2019.09.004
Response: To address the reviewer's concern, we had cited it (Ref 11) and revised our manuscript (page 4, lines 14-16, 18-19; page 5, line 1-2).
- -Obstructive Sleep Apnea (OSA) syndrome is a respiratory sleep disorder characterized by partial or complete episodes of upper airway collapse with reduction or complete cessation of airflow. Although the connection remains debated, several mechanisms such as intermittent hypoxemia, sleep deprivation, hypercapnia disruption of the hypothalamic-pituitary-adrenal axis have been associated with poor neurocognitive performance. Different treatments have been proposed to treat OSAS patients as continuous positive airway pressure (CPAP), mandibular advancement devices (MAD), surgery; however, the effect on neurocognitive functions is still debated. please cite doi:10.3390/bs11120180
Response: To address the reviewer's concern, we cited it (Ref 8) and revised our manuscript (page 4, lines 5-10; page 5, lines 5-7).
- - apply strobe guidelines to improve the structure
Response: To address the reviewer's concern, strobe guideline was attached.
- -add a flow diagram to clarify the protocol
Response: To address the reviewer's concern, flow diagram of the current study had attached in Figure 1.
- - In an interesting study the pattern of overnight changes in serum and urine AKI markers after apneic episodes during sleep may suggest an increased risk of subclinical AKI in patients with OSA. The CPAP therapy is not protective against AKI, but may reduce some of its risk factors, including high blood pressure and endothelial damage. please discuss and cite doi:10.17219/acem/123356
Response: To address the reviewer's concern, we cited it (Ref 19) and revised our manuscript (page 13, lines 3-5, 12-14).

Reviewer 2 Report
It is a fascinating manuscript that surgical treatment of OSA reduces the risk of ESRD but increases non-cardiovascular mortality in CKD patients.
I give some comments to improve this manuscript further.
1. Corrections and modifications are necessary for the overall English expression.
2. A clear description of whether the untreated cohort group means just no surgical treatment or no other treatment such as weight control or CPAP at all is required.
3. I wonder if there is a reason or rationale for dividing the follow-up interval into three months.
4. More detailed descriptions are needed in the Discussion section of this paper.
-It may be because the group that selected surgical treatment may have more severe sleep apnea than the group that not-selected surgical treatment. Still, the intergroup analysis of the severity of OSA is not presented in this manuscript.
-If there is a difference in OSA Severity between the two groups, sufficient analysis and descriptions for reflecting the results are required.
-Surgical treatment can also be done in several ways. There will be various surgical methods, including UPPP and tonsil surgery, nasal surgery, or tongue base surgery. The difference in surgical treatment effect may have influenced the overall surgical treatment effect analysis.
-It is possible that patients who underwent surgery were more willing to treat themselves and were more likely to use CPAP. I think that the difference in the incidence of ESRD due to the difference in CPAP usage cannot be ignored in this study.
Author Response
- Corrections and modifications are necessary for the overall English expression.
Response: To address the reviewer's concern, we received an English editing service and attached a certificate of English editing.
- A clear description of whether the untreated cohort group means just no surgical treatment or no other treatment such as weight control or CPAP at all is required.
Response: The untreated cohort group means just no surgical treatment for OSA, which had shown in the page 7, lines 7-8 of our manuscript. Other treatment such as weight control or CPAP could not be assessed in the NHI claims data, which may exist both in the treated and untreated groups. Now, we mentioned this point in the Method (page 7, lines 9-11).
- I wonder if there is a reason or rationale for dividing the follow-up interval into three months.
Response: The cut point for the follow-up interval of surgical outcomes was not standardized. One research assessed them as short-term (≤14 days), intermediate (15-60 days), and long-term (61-183 days) intervals (Ref 29). But, we assessed the post-operative mortality as short term (≤3 months) and long term (>3 months) in our study. Now, we mentioned this point in the Discussion (page 17, lines 1-4).
- More detailed descriptions are needed in the Discussion section of this paper.
-It may be because the group that selected surgical treatment may have more severe sleep apnea than the group that not-selected surgical treatment. Still, the intergroup analysis of the severity of OSA is not presented in this manuscript.
-If there is a difference in OSA Severity between the two groups, sufficient analysis and descriptions for reflecting the results are required.
Response of both questions: Now, we mentioned and discussed in detail these weak points in the Discussion (page 15, lines 16-19; page 16, lines 1-8). The severity of OSA could not be assessed in the NHI claims data. Thus, the intergroup analysis of the severity of OSA was not presented in the current study. However, these unrecorded factors might violate the results equally in both cohorts. Thus, we supposed that our results would be similar as we reported. Second, the indication for surgical intervention was generally based on the willingness of OSA patients, but not by a randomized method or on the severity of OSA itself. We supposed that the severity of OSA in the treated and untreated groups was similar. Although the severity of OSA in the treated group tended to be more severe compared to the untreated group, its calculated incidence of ESRD was lower. Thus, these findings indeed reflected that surgical treatment for OSA in CKD patients could significantly reduce the incidence of ESRD in CKD patients.
- -Surgical treatment can also be done in several ways. There will be various surgical methods, including UPPP and tonsil surgery, nasal surgery, or tongue base surgery. The difference in surgical treatment effect may have influenced the overall surgical treatment effect analysis.
Response: The surgical methods for OSA could be assessed in the NHI claims data, but we had treated them collectively as one surgical group due to multilevel surgery was performed frequently in modern medicine. Now, we mentioned it in the Method (page 7, lines 9-11) and listed this weak point in the Discussion (page 16, lines 8-11).
6.-It is possible that patients who underwent surgery were more willing to treat themselves and were more likely to use CPAP. I think that the difference in the incidence of ESRD due to the difference in CPAP usage cannot be ignored in this study.
Response: Yes, we could not exclude this possibility. Now, we mentioned it in the Method (page 7, lines 9-11) and listed this weak point in the Discussion (page 15, lines 17-19; page 16, lines 11-19).

Reviewer 3 Report
Dear authors,
I’m very thankful for your work. OSA is common among CKD patients and it could worsen blood pressure control. Your study is conducted on a large number of patients with a notable follow-up. Anyway there are some critical issues:
1. An extensive revision in English style is require. Some sentences are very long and hard to understand. For example: “To prevent immortal bias, we used the incident user (exposed surgical treatment for OSA) design with follow-up for each treated patient beginning on the date of first surgical treatment for OSA and the matching method in which matched untreated patients must be alive at the time when surgical treatment for OSA commenced, and in this situation, cohort entry became the date of first surgical treatment for OSA, and any time between CKD inception date and first surgical treatment for OSA was not counted for either group”
2. CKD definition is inaccurate. Maybe it’s possible to infer from the database which stage of CKD is. This data is not informative.
3. There are several mistakes in tab 1. It’s unreadable.
4. Why do you consider ACE/ARB confounding drug? They are widely used in CKD. I’ m surprised how little is the number of patients treated with ACE/ARB.
5. Patients untreated received CPAP.
6. How do you explain the higher overall mortality and non cardiovascular mortality in treated cohort?
Author Response
Response to Reviewer 3
- An extensive revision in English style is require. Some sentences are very long and hard to understand. For example: “To prevent immortal bias, we used the incident user(exposed surgical treatment for OSA) design with follow-up for each treated patient beginning on the date of first surgical treatment for OSA and the matching method in which matched untreated patients must be alive at the time when surgical treatment for OSA commenced, and in this situation, cohort entry became the date of first surgical treatment for OSA, and any time between CKD inception date and first surgical treatment for OSA was not counted for either group”
Response: To address the reviewer's concern, we attached a certificate of English editing and corrected this sentence in our revised manuscript (page 7, lines 18-19; page 8, 1-8).
- CKD definition is inaccurate. Maybe it’s possible to infer from the database which stage of CKD is. This data is not informative.
Response: The Taiwan's NHI program adopts ICD-9/10 diagnostic codes to identify diseases (page 6, lines 3-5), including CKD (page 6, lines 13-14). The NHI claims data lack information on laboratory data (e.g., serum creatinine). Thus, we could not assess the exact CKD stage (page 6, lines 15-16) and further address the association between CKD stage and the effect of surgical treatments for OSA. The method of adopting ICD diagnostic codes to identify CKD patients in Taiwan has been used in previous high-impact publications [Ref: 14-17]. To address the reviewer's concern, we listed it as a limitation and revised our manuscript (page15, lines 11-16).
- There are several mistakes in tab 1. It’s unreadable.
Response: To address the reviewer's concern, we revised the table 1.
- Why do you consider ACE/ARB confounding drug? They are widely used in CKD. I’m surprised how little is the number of patients treated with ACE/ARB.
Response: ACEI/ARB was considered a confounding drug in CKD research (Ref: 14, 17) because of their renoprotection from CKD progression (page 8, line 9; page 9, line 1). ACEI/ARB prescriptions counted for only 1 year prior to the 1st CKD diagnoses, not thereafter. In our data, 25% of CKD patients diagnosed with SA received ACEI/ARB prescriptions, while only 20% for those with treated SA. Since our target group was CKD patients with treated SA, the propensity score matching scaled down the percent ACEI/ARB prescriptions. Our data showed that CKD+treated SA counted only one thirtieth of CKD+SA which counted only one tenth of all CKD. Indeed a non-randomly sampled minority does not necessarily represent the background population.
- Patients untreated received CPAP.
Response: Yes, some CKD patients with OSA in the non-surgical group might receive CPAP therapy. However, the conclusion of this study would not differ. On the contrary, the effect of surgical treatment for OSA in CKD patients would be more significant due to some subjects in the non-surgical group was not absolutely untreated. Now, we added this point in the Discussion (page 16, lines 11-19).
- How do you explain the higher overall mortality and non cardiovascular mortality in treated cohort?
Response: To address the reviewer's concern, we cited two references (Ref: 29, 30) and revised our manuscript (page 15, lines 1-9).

Round 2
Reviewer 2 Report
There are various limitations mentioned in this manuscript. However, there are many improvements over the previously submitted manuscript.
Reviewer 3 Report
Dear authors, I appreciate your revision agreeing to my remarks. Anyway I found some methodological issue which make weak your conclusions despite of the huge amount of data available.